# The creative interplay between hand gestures, convergent thinking, and mental imagery

**Gyulten Hyusein**[ID]*, **Tilbe Göksun**[ID]

Department of Psychology, Koç University, Sarıyer, Istanbul, Turkey

* ghyusein19@ku.edu.tr

**Data Availability Statement:** The data are available at the Open Science Framework: https://osf.io/qmhvc/?view_only=5a7a8c3e940a46f0ba6f036e3d55a908.

## Abstract

Using hand gestures benefits children's divergent thinking and enhances verbal improvisation in adults. In the present study, we asked whether gestures were also associated with convergent thinking by activating individuals' verbal lexicon and maintaining their visuospatial imagery. We tested young adults on verbal and visual convergent thinking, controlling for their mental imagery skills. Results showed that gestures and mental imagery skills play a role in verbal but not visual convergent thinking. Regardless of whether gestures were spontaneous or encouraged, we found a negative association between overall gesture frequency and verbal convergent thinking or individuals with low mental imagery, and a positive association for individuals with high mental imagery. Representational gestures benefited verbal convergent thinking for everyone except those who had low mental imagery and no experience with the task. Performing beat gestures hampered verbal convergent thinking in people with lower mental imagery capacity and helped those who had high mental imagery and previous experience with the task. We also found that gesturing can benefit people with lower verbal abilities on verbal convergent thinking, however, high spatial imagery abilities were required for gestures to boost verbal convergent thinking. The current study adds a new dimension to both the embodied creativity literature and the kaleidoscope of individual differences in gesture research.

## Introduction

Creativity is one of the most sought-after skills of the 21$^{st}$ century and it is becoming even more important with the rapidly advancing and changing technological environment [1–3]. Creative thinking can be described as a valuable problem-solving skill that enables us to look from many different perspectives at a problem and generate the most appropriate solution, but also as the ability to produce an aesthetically satisfying artwork that is both original and appropriate [4,5]. In recent years, there has been a growing body of research focusing on the potential value of spontaneous and encouraged motions, such as walking [6], enacting metaphors with the whole body or with the hands [2] or gesturing [7,8] on creativity. Most of those studies focused on divergent thinking (generating many different ideas) and no study investigated the effects of gesturing on convergent thinking (i.e., finding the best possible solution to a problem).

**Funding:** This work was supported by a James S. McDonnell Foundation Scholar Award (Grant no: https://doi.org/10.37717/220020510) to Tilbe Göksun. The funders had no role in study design, data collection and analysis, decision to publish, or preparation of the manuscript.

**Competing interests:** The authors have declared that no competing interests exist.

Gestures are a special case of action or movement because they bear both communicative and cognitive off-loading functions [9,10]. In addition to that, gestures reflect mental representations, which sometimes are not expressed in speech, but are also activators of mental images themselves [11–13]. The reciprocal manner in which gestures contribute to thinking and speaking can be advantageous for the formation of new creative ideas and reaching insights [14].

The aim of the current study is to investigate whether gesturing, both encouraged and spontaneous, would be beneficial for connecting remote associations, which is a type of convergent thinking, and identify the role of mental imagery in this process. This is a topic of relative significance because it will not only help us find out more about the self-oriented functions of gestures for thinking and problem-solving but also because it might have practical implications for developing methods to boost creative thinking skills.

## What is creative thinking?

The Partnership for 21st Century Learning pointed out four skills and competencies essential for teaching and learning, named as the 4Cs: *communication*, *collaboration*, *critical thinking*, and *creativity* [3]. The last one–*creativity*—is generally defined as the phenomenon of formation of something abstract, such as an idea, or an object, e.g., an innovative product, which should be both original and effective [5,15–17]. Even though both cognitive scientists and artists have questioned the originality and effectiveness criteria for creativity [18,19], this definition is still the most widely accepted one. Another aspect of creativity that is worth considering is that creative thinking is not solely in the realm of highly imaginative children or artists, but it is a skill that every person can develop [20].

In general, two types of creative thinking are differentiated in the literature: *divergent* and *convergent thinking*. While divergent thinking is defined as the ability to generate various new and appropriate ideas or solutions to a problem, convergent thinking is related to having an insight about a problem after a more constrained mental search for the best possible answer [21,22]. Even though divergent and convergent thinking may seem as opposing concepts, researchers usually agree that creativity is a cycle of both and occurs in combination with other higher-order cognitive processes, such as mental imagery, analogical reasoning, and metaphorical thinking [3,23].

A prevalent debate in creativity research is whether creative thinking is domain-specific or rather, domain-general. Some researchers support the domain-specificity argument or the idea that being creative in one area, e.g., artistic creativity, does not make one creative in other areas, e.g., scholarly or everyday creativity, because creativity has different origins for different fields [24–26]. However, studies also show that creativity can be domain-general and traits, skills, or abilities in one domain can also be translated into other areas of creativity [27–29]. A more reconciling and recently, widely supported view is that creativity can be both domain-specific and domain-general [30,31]. By adopting the attitudes used in its different domains we can foster its domain-generality [32] and thus, improve both divergent and convergent thinking skills.

## Gestures and thought

Gestures are spontaneous hand movements that occur when one is speaking or thinking [33–35]. McNeill [36] classified gestures into four categories: *iconic*–to convey aspects of objects or actions through their form (e.g., moving your index and middle fingers back and forth to indicate walking); *metaphoric*–to represent abstract ideas (e.g., outlining the space in front of the body in reference to time); *beat*–rhythmic hand movements to give emphasis to the discourse

structure; and *deictic*–pointing with the hand or the fingers to refer to objects, people or locations in the immediate environment. Since iconic, metaphoric, and deictic gestures bear a semantic meaning that is usually overlapping with the verbal message, they are called *representational* gestures, while beat gestures are classified as *non-representational*.

Gestures have a communicative role in facilitating speaker's speech and listener's comprehension [9,13,37,38], but there is also growing evidence that gestures convey cognitive and mental states (e.g., [39,40]). They act as an off-loading platform for cognition [13] and have the power to influence thinking and speaking by activating mental images and simulating actions [10–12]. Goldin-Meadow and Beilock [10] argued that gestures are more powerful than mere actions because while actions' main purpose is to influence the world more directly, gestures have the potential to influence thinking. When gesturing, one needs to generate an internal mental representation of an object or of an abstract idea and 'off-load' it to the environment in the form of gestures. Interestingly, gestures are not only reflections of one's thinking, but they can also feedback and change one's thoughts. Moreover, what is conveyed in gestures is sometimes not expressed in words, thus, gestures are also reflections of ideas or mental representations that the speaker might not be explicitly aware of. For example, when encouraged to gesture while explaining answers to a math problem, children activated ideas that were present in their repertoire but unexpressed in their speech [41].

The idea of gestures' role as an off-loading medium for cognition has also been suggested by the *Gesture-for-Conceptualisation Hypothesis* proposed by Kita et al. [13]. According to this framework, gestures, and more specifically representational gestures, aid thinking and speaking by *activating*, *manipulating*, *packaging*, and *exploring* spatio-motoric information. In a similar fashion to Goldin-Meadow and Beilock [10], Kita and colleagues [13] distinguish gestures from practical actions, such as object manipulation, by stating that gestures have a self-oriented function in representing and schematising information. The *Gesture as Simulated Action* (GSA) framework [11,12] further emphasises the image-based activation of representational gestures. The framework proposes that gestures occur as a result of the activation of motor and visual areas in the speakers' brain in response to mental images that are then expressed in gestures while speaking. This is how mental images so naturally lead to representational gestures, i.e., iconic and metaphoric gestures.

Even though these hypotheses highlight the role of representational gestures in thinking, studies have found that beat gestures may also have beneficial functions for word retrieval (e.g., [42,43]). Lucero et al. [42] instructed participants to produce beat or iconic gestures while coming up with a word after seeing its definition. Those who produced beat gestures were faster to elicit the target word compared to the iconic or a baseline (no-gesture) condition. Similarly, So et al. [43] found that encoding beat gestures aided memory recall for adults but not for children. Encoding iconic gestures, on the other hand, was beneficial for both children and adults. The authors speculated that children might not be sensitive to the meta-cognitive aspects reflected in beat gestures, but adults were. These are intriguing results because beat gestures do not render any meaning while iconic gestures bear semantic representations. Hence, these findings show that not only representational but also non-representational gestures might facilitate thinking and speaking.

## Why would gestures boost creative thinking?

In his *Action-Cognition Transduction* hypothesis, Nathan [14] argues that the function of gestures is to create new ideas–both insights and inferences. His central claim is that cognitive and motor functions operate in a reciprocal manner and form a coherent system. Furthermore, he states that gestures impact equally *System 1* intuitive behaviours and *System 2*

deliberate behaviours through a process of transduction and facilitate the generation of new ideas both consciously, e.g., by inferences and analysis, and unconsciously, e.g., leading to unconscious insights or fostering the articulation of latent ideas as in Broaders et al.'s [41] study. This account is also supported by mounting evidence about the potential value of embodied experimental manipulations, including gestures, on creative idea generation and problem-solving. Previous studies highlight the positive impact of gesturing on divergent thinking [7,8,44,45], and other work suggests embodied cognition's benefits for convergent thinking [1,2,46,47]. The latter, however, rarely included the effects of gestures on convergent thinking.

**Gestures and divergent thinking.** Both children and adults who either naturally gestured more or were encouraged to gesture showed an increase in their improvisation skills [45], idea generation ability [7,8], and creative story retelling capacity [44]. Lewis et al. [45] found that gesturing spontaneously enhanced verbal improvisation measured by fluency, originality, elaboration, and flexibility of ideas. Improvising elicited a higher number of iconic and deictic gestures compared to everyday speech. The higher the quality of improvisations, the higher the rate of iconic and beat gestures was, while weaker improvisors produced more abstract gestures referring to the improvisation object–a 'blueblepip on their shoulder.' However, this study did not include metaphoric gestures in their coding procedure. Metaphoric gestures might be important in the formation of creative thought, which was supported by a group brain-storming study conducted by Liao and Wang [8]. Here, the authors coded iconic, metaphoric, pointing, and other (not classified into the previous three categories) gestures and found that only metaphoric gestures influenced both self's and partner's idea generation. A limitation of this study, nevertheless, could be that beat gestures were not coded separately so we do not know if they also had a beneficial role in idea generation as they did in Lewis et al.'s [45] improvisation task.

As it comes to the effects of gestures on children's divergent thinking, Laurent et al. [44] showed that children who spontaneously gestured more, told longer, and more creative stories with creativity measured as the number of low-frequency words used and scene reordering when retelling a previously heard story. Earlier, Kirk and Lewis [7] also showed that encouraging gestures increased fluency but not originality or flexibility of ideas in children on the Alternative Uses Task (AUT; [21]). Interestingly, restricting them from gesturing did not influence children's idea generation, which the authors attributed to children's ability to come up with other strategies for idea generation.

**Embodied convergent thinking.** There have not been any studies investigating gestures' effects on convergent thinking. However, research has shown that fluid arm movements, squeezing a ball, or enacting metaphors with hands could improve convergent thinking [1,2,46,47]. For example, Slepian and Ambady [47] showed that embodying the metaphor of fluid thinking by enacting fluid arm movements helped participants connect remote associates, while rigid/nonfluid arm movements did not. Similarly, Leung et al. [2] found that sitting outside a box to represent the metaphor of "thinking outside the box" facilitated connections on the Remote Associates Test (RAT; [22]). Furthermore, making hand movements that involved integrating objects, i.e., combining paper from a left-positioned stack with paper from a right stack and placing the combination in the middle, facilitated convergent thinking. While these studies explored the effects of enacting metaphors on convergent thinking, other studies indicated that simple hand contractions can benefit convergent thought. For instance, Goldstein et al. [46] showed that left-hand muscle contractions were more beneficial for convergent thinking compared to right-hand or no contractions at all. This effect was explained by the importance of the right hemisphere in creative thinking, as this was the hemisphere activated by the movement of the contralateral limb. In addition, Kim [1] found that squeezing a

hard ball but not a soft ball with the non-dominant hand was more beneficial for convergent thinking when solving remote associate problems. Even though the authors did not mention what proportion of their sample was left or right-handed, considering the general population, it can be assumed that most participants were right-handed. Hence, they mostly squeezed the ball in their left hand, which would activate their right hemisphere. Whether the same effect is true for gestures executed with the left hand is a question for further research.

The empirical research presented thus far underscores the positive effects of hand movements on creative thought. Gesture-formation theories propose that gestures' beneficial role for thinking might stem from gestures' role in the activation of mental images [10–12]. Similarly, both theoretical frameworks of creativity [48,49] and empirical studies [50–53] have identified the importance of mental imagery in the creative thinking processes. However, most of the creativity measurements of these studies involved divergent thinking tasks [7,8,50,53] and convergent thinking has been neglected. In the next section, we will go into more details on the role of mental imagery in the gesture-creativity relationship.

## Is mental imagery important for the gesture-convergent thinking interplay?

The most discussed type of mental imagery is *visual* mental imagery, i.e., "seeing in the mind's eye" or "visualising," which is the ability to generate picture-like representations (mental images) in one's mind, maintain their vividness and be able to perform manipulations with those images, such as mental rotations [54,55]. Mental imagery plays a significant role both in creative thinking and in the process of gesture-formation.

Mental imagery is an integral part of creative thinking, supporting the access to thoughts and images that facilitate the development of new ideas and creative achievements [56]. According to the creative cognition approach, which is based on the 'Geneplore' model [48], creativity emerges through generative (e.g., mental synthesis) and exploratory (e.g, conceptual interpretation) processes and mental imagery has a pivotal role in guiding creative ideas. Empirical studies support this view [50,57–59]. For example, Vellera and Gavard-Perret [58] reported that artists and inventors had more vivid mental images than 'ordinary' individuals and in a second study, they found that higher self-reported imagery scores were associated with better performance on a creativity task. Moreover, both domain-general and domain-specific creativity in undergraduate dance students were enhanced after mental imagery training [50]. Another recent neuroimaging study argued that mental imagery might have a mediating role in the relationship between episodic memory and divergent thinking [59]. Mental imagery may help individuals recombine their episodic memory experiences with new mental images, which in turn promoted divergent thinking.

The Gesture-for-Conceptualisation hypothesis and the Gesture as Simulated Action framework are based on the proposition that gestures are activated by mental images, which has also been supported by empirical research. For instance, in a study conducted both with younger (mean age of 21) and older (mean age of 66) adults, Arslan and Göksun [60] found that it was mental imagery skills but not working memory that predicted the use of representational gestures for both groups. Moreover, Laurent et al. [44] found that pre-schoolers who gestured more told more creative narratives. They speculated that gesturers may be relying upon vivid visuospatial cognitive resources or imagery to a greater extent and hence, show a tendency to incorporate more creativity into their narratives compared to preschoolers who did not spontaneously gesture.

Studies examining the explicit role of mental imagery on creativity generally used divergent thinking tasks to measure creativity. For example, Vellera and Gavard-Perret [58] asked

participants to draw an imagined creature; May et al. [50] used the Abbreviated Torrance Test for Adults (ATTA; [61]); and Zhang et al. [59] asked participants to think of unusual and interesting ways to make a toy more appealing. A topic worth exploring is whether mental imagery is equally related to convergent thinking as it is to divergent thinking and if so, whether gestures would be facilitative in activating those mental images. To account for the complex nature of mental imagery, the phenomenon is generally studied in relation to its four components: *generation*, producing or generating an image, *maintenance*, holding the image in working memory, *inspection*, focusing on and studying specific parts of the image, and *transformation*, transforming or manipulating the image in the mind [62,63]. However, as the recent literature on mental imagery emphasises the need to differentiate between imagery styles, such as *spatial* and *object*, and even a more symbolic and abstract one, i.e., *verbal* [64–66], it is necessary to study individual differences in mental imagery in relation to gestures and creative thinking processes.

## The present study

The aim of the current study is to explore the relationship between gesture use (either *spontaneous* or *encouraged*) and convergent thinking in young adults. Frequency and types of gestures will be examined in relation to solving two convergent thinking tasks–verbal Remote Associate Task (RAT) [22] and visual Remote Associate Task (vRAT) [67]. Moreover, to determine the role of mental imagery skills in the gesture-convergent thinking relationship, we will use the Mental Imagery Test (MIT; [68]). The MIT is a standardised battery assessing maintenance, inspection, generation, and manipulation of different categories of images. In addition to task-measured creative thinking and mental imagery, we will also ask participants to fill out a self-report questionnaire regarding their perceived imagery styles (Object-Spatial Imagery and Verbal Questionnaire; OSIVQ; [69]). Obtaining both experimental and self-report data will help us gain a better understanding of the intricate nature of creative thinking, gesture use, and mental imagery.

As there are large individual differences in gesturing rate and type across individuals [70,71], this study will adopt both a between- and a within-subjects design. One group of participants will be exposed to the verbal and visual RAT tasks without any mention of hand movements, and later solve similar RAT problems while encouraged to gesture. A second group of participants will solve the verbal and visual RAT questions while being encouraged to gesture. The within-subject effects will allow for more power to detect the effect of condition accounting for the individual variations in gesturing, and the between-subject condition will help us partially eliminate the practice effects of the repeated measures design.

First, we hypothesise that people who spontaneously gesture more would be better at connecting remote associates as their gestures might act as triggers of mental images helping them more readily visualise the given information. Second, encouraging gesture use, on one hand, is also expected to improve connecting remote associations by strengthening the underlying mental simulations of images but on the other hand, might deteriorate performance by increasing cognitive load. For example, when asked to gesture, people performed worse on analogical problem-solving tasks and this was especially the case when they gestured about elements that were not relevant to the task's solution [72,73]. Therefore, we expect more relevant gestures, such as representational gestures, to be more beneficial for connecting remote associates in comparison to beat gestures.

Third, we expect participants who have more advanced mental imagery skills as measured by the MIT to use their gestures more frequently both as a reflection of their internal mental images and as feedback for connecting remote associates on the convergent thinking task. As

it comes to self-reported imagery styles (spatial, object, and verbal), we anticipate those who perceive themselves as object visualisers to do better at the visual RAT, and those who see themselves as verbal visualisers to be better at the verbal RAT when their gestures are encouraged or when they naturally gesture more during the tasks. As our convergent thinking measures do not involve any spatial relationships or transformations, we do not expect any correlations between the RAT, either verbal or visual, and the spatial imagery style. However, we expect the MIT to positively correlate with visual and spatial imagery styles as it measures the maintenance, inspection, generation, and manipulation of images.

## Method

### Participants

A total of 90 young adults ($M_{age}$ = 21.4, $SD$ = 2.46; 57 females) took part in the current study. All of them were Turkish native speakers. Forty-eight of the participants were undergraduate students who were recruited through Koç University's subject pool for partial fulfilment of course credit. The rest of the young adults were recruited based on convenience sampling. The data of nine participants were removed from the final analysis due to technical errors and one participant was discarded because they reported having an advanced visual disability during the testing procedure. For the RAT analyses, two participants were later excluded because one of them was not shown the RAT triads on the screen but the experimenter only read them out loud; and the second one was excluded because they were an outlier due to previous research familiarity with the task. Thus, our final sample consisted of 78 participants ($M_{age}$ = 21.3, $SD$ = 2.47; 50 females) for the RAT analyses and 80 participants ($M_{age}$ = 21.3, $SD$ = 2.46; 52 females) for the vRAT analyses.

On recruitment, participants were randomly assigned either to Group 1 ($N$ = 40) where they completed both a gesture-spontaneous and a gesture-encouraged conditions or to Group 2 ($N$ = 40) where they only completed a gesture-encouraged condition. The sample size was determined based on previous studies [7,8]. Ethical approval was granted by Koç University Ethical Committee on Human Research (IRB Protocol Number: 2020.125.IRB3.063).

### Materials

#### Convergent thinking tasks

*Remote Associates Test (RAT; [22]).* The Turkish version of RAT was used to measure verbal convergent thinking [74]. In this task, participants are presented with three unrelated words (e.g., cottage, swiss, cake) and are asked to come up with a fourth word that is a common associate of those three words (cheese). We used 10 triads in each condition. The words were presented on a white screen and were read out loud by the experimenter. Then, participants had 30 seconds per triad to come up with a relevant answer. Scoring was based on the number of correct responses with each correct response scored as 1.

*Visual Remote Associates Test (vRAT; [67]).* This task was used as a measure of visual convergent thinking. Participants were presented with three seemingly unrelated objects (e.g., glove, handle, and pen) and asked to find a common associate that would co-occur with each of the three objects (hand). The objects were presented as pictures on a white screen and participants had 30 seconds per triad to name the common associate. We used 10 triads in each condition. Since this task has not been validated for the Turkish culture, we chose those triads from the original pool of stimuli that would also be relevant for the Turkish context. In addition to that, upon data collection the authors of the study together with three research assistants agreed on alternative valid answers that were counted as correct in addition to the

original answers proposed by Oltețeanu [67]. Scoring was based on the number of correct responses with each correct response scored as 1.

**Imagery measures.** *Mental Imagery Test (MIT; [68]).* This is a battery of eight tasks designed to measure mental imagery skills, namely *generation*, *maintenance*, and *manipulation* of different categories of images. This task was used before in a Turkish sample with Turkish instructions [60]. The task stimuli were presented on a white screen and answers were noted down in the answer sheet by the experimenter during the sessions. A total score of mental imagery was calculated by adding up the scores of the single subsets.

*Object-Spatial Imagery and Verbal Questionnaire (OSIVQ; [69]).* We used the Turkish adaptation of the OSIVQ [75]. OSIVQ is a self-report measure of visual (spatial and object) and verbal cognitive styles. The scale consists of 45 items with 15 items measuring each dimension. Sample items are as follows: "I can easily sketch a blueprint for a building I am familiar with" for spatial cognitive style; "My images are very colourful and bright" for object cognitive style; "My verbal abilities would make a career in language arts relatively easy for me" for verbal cognitive style. Participants rated each statement on a 5-point Likert-type scale with '1' indicating complete disagreement and '5'–complete agreement. Scores for each subset were calculated by averaging the 15 ratings, respectively.

**Procedure.** Experiments were conducted on-line in one-to-one sessions on Zoom. The data were collected by the first author and three trained research assistants. All sessions were video recorded for further coding. After obtaining verbal informed consent from the participant, their position was adjusted so that their upper body and hands could be captured by the camera. However, we did not mention anything about gesture use, particularly before testing Group 1 –gesture-spontaneous condition. We told the participants that the adjustment of the position was made to make the recording consistent across participants. To acquire information about baseline gesture rate, we started the sessions by asking the question "What do you do on a regular day?". After that, we administered the convergent thinking tasks (RAT and vRAT) in a counterbalanced manner. The stimuli, either words or pictures, were displayed by sharing the experimenter's screen.

If the participant was assigned to Group 1, they completed the convergent thinking tasks twice–first in a gesture-spontaneous condition where there was no mention of gestures or hand movements, and after that, in a gesture-encouraged condition where, before each task, we explicitly encouraged them to use their hands while thinking and speaking. If the participant was assigned to Group 2, they only completed the convergent thinking tasks in the gesture-encouraged condition. The reason for having Group 2 was to be able to disentangle learning or practice effects from the effects of gesturing alone. The same stimuli were used for Group 1 and Group 2 in a counterbalanced order across groups and conditions.

Following the convergent thinking tasks, participants completed the MIT. After that, the recording was stopped, and the participant was sent a Qualtrics link redirecting them to a demographic form and the self-report imagery styles measure–OSIVQ. When the participant completed the questionnaires on Qualtrics, they were debriefed and thanked for their time or awarded course credits if they were recruited through the university's subject pool system. Depending on the group the participant was assigned to, the duration of the whole session varied between 50–70 minutes.

**Transcription and coding.** Trial-by-trial speech and gestures for the RAT and vRAT were transcribed and coded on the ELAN language archive software (Version 6.0; 2020) by two trained research assistants. Initially only McNeill type co-speech gestures were coded, i.e, iconic, deictic, metaphoric, and beat [36]. During the coding procedure, the first author and the coders noticed that the participants frequently used palm-revealing gestures (i.e., gestures where the palm rotates up to express uncertainty or the speaker has nothing else to say). Palm-

revealing gestures were originally incorporated in Chu et al.'s [76] coding scheme. Because of their frequent presence in the RAT and the vRAT tasks, we also coded them as they may be meaningful for the current tasks and research questions. The first assistant transcribed and coded speech and gestures for RAT and the second assistant–for vRAT. Later, each of them coded 20% of the other one's data to obtain inter-rater reliability scores. To resolve the inconsistencies, the first author together with another trained assistant recoded the gestures that seemed to be miscoded in the original coding procedure. The reliability coding procedure was repeated for the corrected gesture coding until the interrater agreement was substantial ($\kappa$ = .751, $p$ < .001). Discrepancies in identifying and classifying gestures into the aforementioned categories were resolved by further discussion to reach a full agreement. We calculated gesture frequency for each gesture type by dividing the number of gestures produced by the number of words spoken in each trial. We also calculated representational gesture frequency by dividing the total number of representational gestures (iconic, deictic, and metaphoric) by the number of words spoken in each trial.

The data are available at the Open Science Framework: https://osf.io/qmhvc/?view_only=5a7a8c3e940a46f0ba6f036e3d55a908.

## Results

We used linear (lmer) and generalised binomial (glmer) linear mixed-effects modelling to test our hypotheses. We included random intercepts for Subject and Item for our between-group analyses (gesture-encouraged condition group 1 (GE1) and gesture-encouraged condition group 1 (GE2)) and a random intercept for Item and a random slope of Condition by Subject for our within-group analyses (gesture-spontaneous condition group 1 (GS1) and gesture-encouraged condition group 1 (GE1)). The mixed-effects approach allowed us to account for the random variability due to having different participants and different items and adding random slopes into our models allowed us to include Condition as a within-subject variable. We used the lme4 package (version 1.1.17; [77]) in R [78] with the optimizer *bobyqa* and scaling of continuous predictors to prevent non-convergence [77,79]. Significance was tested with the Anova Type III Sum of Squares of the lmerTest package. To understand two- and three-way interaction effects we used the *sim_slopes* function from the *jtools* package [80].

### Preliminary results

Table 1 presents the mean and standard deviation values for age, the verbal and the visual RAT, the mental imagery test scores, and gesture frequency rates during the verbal and the visual RAT across groups and conditions (see the S1 File for a detailed description and visualisation of gesture types and frequencies across groups and conditions).

### RAT (verbal) within-group (GS1 and GE1) and between-group (GE1 and GE2) analyses

Hypothesis 1: People who naturally gesture more should do better on the RAT because of high mental imagery.

To test this hypothesis, we fit a model where RAT scores were the outcome variable, total gesture frequency, and mental imagery (MIT) scores were fixed effects, and subject and item were random effects. We found a significant two-way interaction between gesture frequency and MIT (See Table 2). Simple slope analyses showed that for people with mean and above the mean MIT scores, there was a positive association between spontaneous gesture frequency and RAT scores; for mean MIT: $\beta$ = 0.52, $SE$ = 0.21, $p$ = .01; for MIT above the mean: $\beta$ = 1.15, $SE$ = 0.30, $p$ < .001; for MIT below the mean: $\beta$ = - 0.11, $SE$ = 0.24, $p$ = .63.

**Table 1. Mean (*M*) and standard deviation (*SD*) values of age, RAT, vRAT, and MIT scores and gesture frequency across groups.**

| | Groups | | | | | |
| --- | --- | --- | --- | --- | --- | --- |
| | Gesture-Spontaneous Group 1 (GS1) | | Gesture-Encouraged Group 1 (GE1) | | Gesture-Encouraged Group 2 (GE2) | |
| | *M* | *SD* | *M* | *SD* | *M* | *SD* |
| Age | 21.6 | 2.74 | 21.6 | 2.74 | 21.0 | 2.14 |
| RAT Score | 5.82 | 1.78 | 6.61 | 2.01 | 6.10 | 2.22 |
| vRAT Score | 6.25 | 1.77 | 6.33 | 1.66 | 5.88 | 2.03 |
| MIT Score (RAT sample) | 69.6 | 7.75 | 69.6 | 7.75 | 68.5 | 7.27 |
| MIT Score (vRAT sample) | 69.5 | 7.67 | 69.5 | 7.67 | 68.8 | 7.54 |
| Overall Gesture Frequency during RAT | 0.08 | 0.21 | 0.20 | 0.41 | 0.19 | 0.37 |
| Overall Gesture Frequency during vRAT | 0.09 | 0.31 | 0.18 | 0.40 | 0.17 | 0.32 |
| Representational Gesture Frequency during RAT | 0.01 | 0.06 | 0.13 | 0.37 | 0.03 | 0.12 |
| Representational Gesture Frequency during vRAT | 0.02 | 0.11 | 0.09 | 0.26 | 0.02 | 0.12 |
| Beat Gesture Frequency during RAT | 0.03 | 0.14 | 0.09 | 0.30 | 0.05 | 0.17 |
| Beat Gesture Frequency during vRAT | 0.05 | 0.21 | 0.15 | 0.34 | 0.11 | 0.26 |

Hypothesis 2: Encouraging gesture use improves (or hampers) RAT performance and if it improves performance, this is specifically true for representational gestures.

*Overall Gesture Use*: To test the hypothesis that encouraging gesture use would improve or impede performance on the RAT, firstly, we fit a model with RAT scores as the outcome variable, total gesture frequency, condition, and MIT as the fixed effects. Condition (spontaneous vs. encouraged) was added as a random slope into participants (subjects) and item was added as a random intercept. This was our within-subjects effect model. The model revealed a significant interaction between gesture frequency and MIT, but no effect of condition (see Table 3). For participants with MIT below the mean, there was a negative association between gesture frequency and RAT scores, $\beta = -0.40$, $SE = 0.14$, $p < .001$. On the other hand, for participants

**Table 2. Model summary for Hypothesis 1.**

| RAT | | | vRAT | |
| --- | --- | --- | --- | --- |
| | Coefficient | *SE* | Coefficient | *SE* |
| FIXED EFFECTS | | | | |
| Intercept | 1.14[†] | .60 | .83* | .38 |
| Gesture Frequency | .52* | .21 | .11 | .21 |
| MIT | .37 | .28 | -0.4 | .17 |
| Gesture Frequency*MIT | -.63*** | .16 | -.35 | .27 |
| | Variance | *SD* | Variance | *SD* |
| RANDOM EFFECTS | | | | |
| INTERCEPTS | | | | |
| Subject | 2.64 | 1.62 | .10 | .32 |
| Item | 5.52 | 2.36 | 2.30 | 1.52 |

*Note*: MIT–Mental Imagery Test.

Significance codes =

***$p < .001$

**$p < .01$

*$p < .05$

†$p < .1$.

**Table 3. Model summary for Hypothesis 2.**

| | RAT | | | | vRAT | | | |
|---|---|---|---|---|---|---|---|---|
| | within-subjects (Group 1) | | between-subjects (Group1&Group2) | | within-subjects (Group 1) | | between-subjects (Group1&Group2) | |
| | Coefficient | SE | Coefficient | SE | Coefficient | SE | Coefficient | SE |
| FIXED EFFECTS | | | | | | | | |
| Intercept | 1.79* | .52 | 1.21* | .54 | .86* | .38 | .76* | .38 |
| Gesture Frequency | .18 | .11 | .19* | .09 | .08 | .12 | .18† | .11 |
| Condition | -.26 | .29 | | | -.08 | .19 | | |
| Group | | | -.21 | .39 | | | -.34 | .24 |
| MIT | .38† | .23 | .60** | .20 | .11 | .24 | .11 | .12 |
| Gesture*Condition | .33 | .22 | | | .11 | .24 | | |
| Gesture*Group | | | .30† | .18 | | | .36† | .22 |
| Gesture*MIT | .57*** | .10 | .71*** | .10 | -.12 | .14 | .01 | .12 |
| Condition*MIT | -.18 | .29 | | | -.16 | .20 | | |
| Group*MIT | | | .26 | .40 | | | -.09 | .24 |
| Gesture*Condition*MIT | -.06 | .19 | | | -.53† | .29 | | |
| Gesture*Group*MIT | | | .20 | .21 | | | -.13 | .25 |
| | Variance | SD | Variance | SD | Variance | SD | Variance | SD |
| RANDOM EFFECTS | | | | | | | | |
| INTERCEPTS | | | | | | | | |
| Subject | 2.76 | 1.67 | 2.63 | 1.62 | .30 | .55 | .46 | .68 |
| Item | 4.40 | 2.10 | 4.96 | 2.23 | 2.47 | 1.57 | 2.56 | 1.60 |
| SLOPES | | | | | | | | |
| Condition|Subject | 2.69 | 1.64 | | | .01 | .07 | | |

*Note*: MIT–Mental Imagery Test.

Significance codes =

***$p < .001$

**$p < .01$

*$p < .05$

†$p < .1$.

with MIT above the mean, there was a positive association between gesture frequency and RAT scores, $\beta = 0.74$, $SE = 0.15$, $p < .001$ (see Fig 1A).

The same hypothesis was also tested with the gesture-encouraged conditions only (between-group model). Gesture frequency, group (GE1 vs. GE2), and MIT were added as fixed effects, while item and subject were added as random intercepts. Similar to our within-group analyses, here, there was a two-way interaction between gesture frequency and MIT, however, the three-way interaction between gesture frequency, group, and MIT score was not significant (see Table 4). Irrespective of group, i.e., whether participants attended a gesture-spontaneous condition before the gesture-encouraged one or not, there was an effect of MIT skills on the relationship between gestures and RAT scores. For those with MIT below the mean, there was a negative relationship between gesture frequency and RAT, $\beta = - 0.55$, $SE = 0.12$, $p < .001$, while for those with MIT above the mean, the relationship between gesture frequency and RAT was a positive one, $\beta = 0.86$, $SE = 0.15$, $p < .001$ (see Fig 1A). In both the within- and between-analyses, for people with mean MIT scores, there was no effect of MIT on the relationship between gesture frequency and RAT; within-effects: $\beta = 0.17$, $SE = 0.11$, $p = .12$; between-effects: $\beta = 0.16$, $SE = 0.09$, $p = .08$.

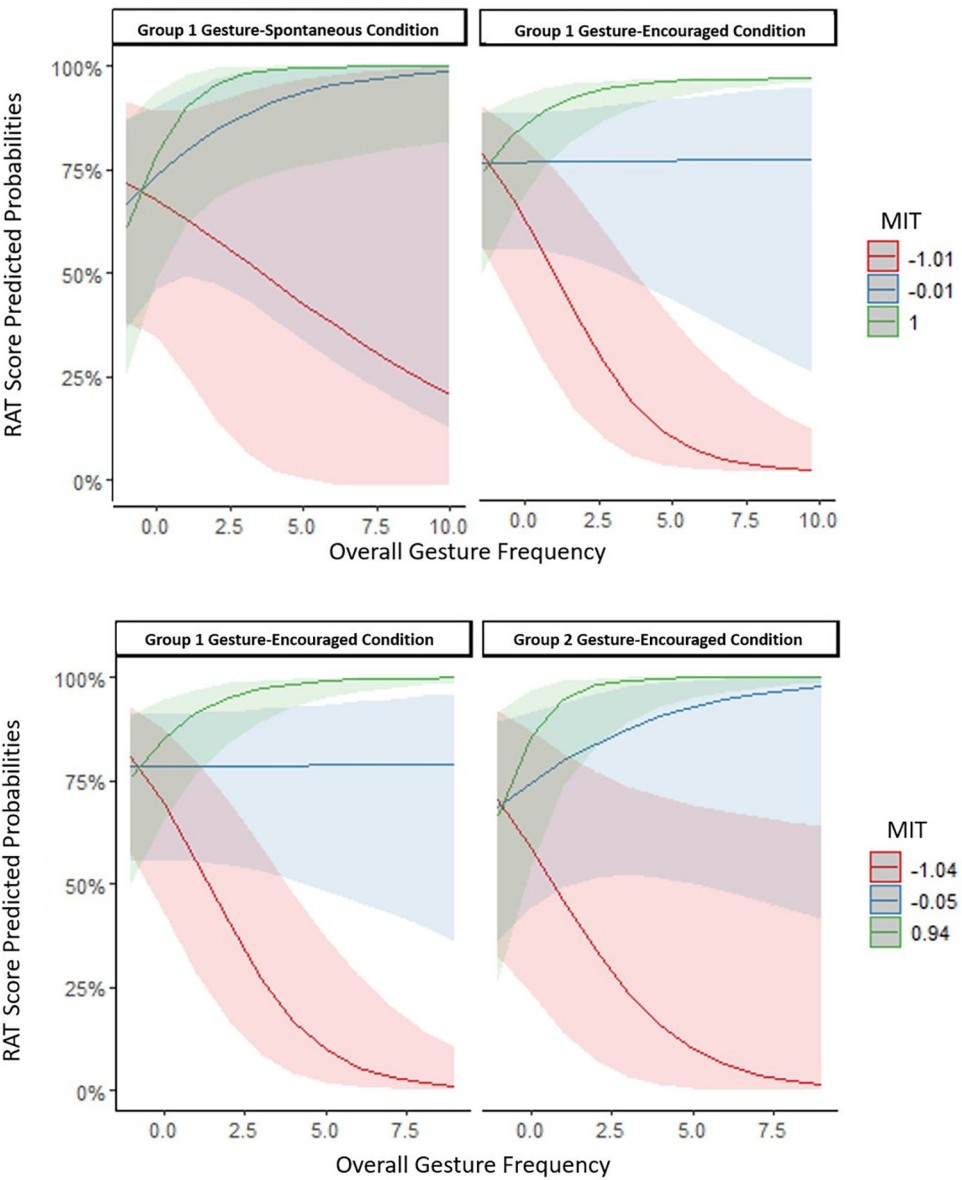

**Fig 1. The relationship between predicted probabilities of RAT scores and scaled overall gesture frequency as a factor of scaled MIT scores (mean, below and above the mean) for the gesture-spontaneous and the gesture-encouraged condition of Group 1 and the gesture-encouraged condition of Group 2. A:** within-subject effects; **B:** between-subject effects.

Hypothesis 2 continued: The role of gesture types in convergent thinking and its interaction with mental imagery.

Next, we fit separate models for gesture types, i.e., beat, representational (iconic, meta-phoric, pointing/deictic), and palm-revealing gestures. In all models, when testing the within-subjects effects, the specific gesture type, MIT scores, and condition were entered as fixed factors, while item was entered as a random intercept, and condition was added as a random slope of subject. When testing for between-subjects effects, the specific gesture type, MIT scores, and condition were entered as fixed factors, and item and subject were random intercepts. Here, we will focus on beat and representational gestures. Results for the individual

**Table 4. Model summary for Hypothesis 2—Beat gestures.**

| | RAT | | | |
| | within-subjects (Group 1) | | between-subjects (Group 1 & Group 2) | |
| | Coefficient | SE | Coefficient | SE |
|---|---|---|---|---|
| FIXED EFFECTS | | | | |
| Intercept | 1.23* | .54 | 1.25* | .55 |
| Beat Frequency | -.28** | .10 | -.24* | .10 |
| Condition | -.50† | .29 | | |
| Group | | | -.36 | .41 |
| MIT | .39 | .25 | .58** | .21 |
| Beat*Condition | .30 | .21 | | |
| Beat*Group | | | .31† | .19 |
| Beat*MIT | .50*** | .11 | .72*** | .13 |
| Condition*MIT | -.46 | .29 | | |
| Group*MIT | | | -.01 | .41 |
| Beat*Condition*MIT | -.83*** | .22 | | |
| Beat*Group*MIT | | | -.68** | .25 |
| | Variance | SD | Variance | SD |
| RANDOM EFFECTS | | | | |
| INTERCEPTS | | | | |
| Subject | 3.54 | 1.88 | 2.91 | 1.71 |
| Item | 4.48 | 2.12 | 5.17 | 2.27 |
| SLOPES | | | | |
| Condition/Subject | 2.84 | 1.68 | | |

*Note*: MIT–Mental Imagery Test.

Significance codes =

***$p < .001$

**$p < .01$

*$p < .05$

†$p < .1$.

representational gesture types, i.e., iconic, metaphoric and pointing/deictic gestures, and palm-revealing gestures can be found in the S1 File.

*Beat gestures*: In the within-effects model including beat gestures, the three-way interaction between beat gesture frequency, condition, and MIT was significant (see Table 3). Simple slope analyses revealed that for people who are below the mean on the MIT, when encouraged to gesture, there was a negative association between beat gesture frequency and RAT scores, $\beta$ = - 3.15, $SE = 0.67$, $p < .001$. However, for those who were above the mean on MIT, when encouraged to gesture, beat gestures were a significant positive predictor of RAT scores, $\beta$ = 2.25, $SE = 0.61$, $p < .001$. No such trends were detected for beat gestures in the gesture-spontaneous condition or for people with mean MIT scores (see Fig 2A).

In the between-effects model including beat gestures, the three-way interaction between beat gesture frequency, group, and MIT significantly improved the model, $\chi2$ (1) = 7.27, $p$ = .007. Simple slope analyses showed that in GE1 (Group 1's gesture-encouraged condition), for people with mean and below the mean MIT skills, the more beat gestures they executed when encouraged to do so, the worse they did on the RAT, mean MIT: $\beta$ = -0.44, $SE = 0.12$, $p < .001$; below the mean MIT: $\beta$ = -1.44, $SE = 0.26$, $p < .001$. However, in the same group (GE1), people

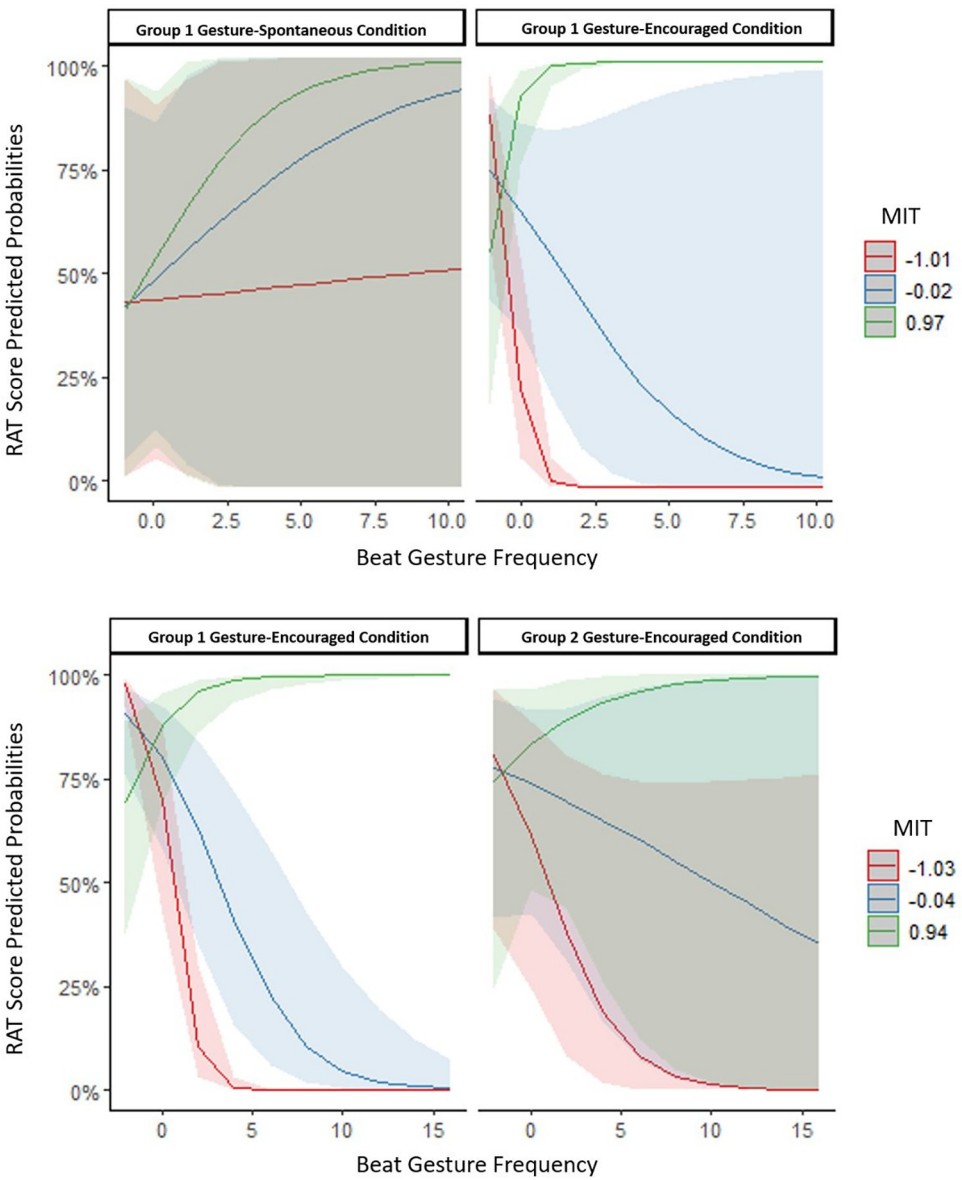

**Fig 2. The relationship between predicted probabilities of RAT scores and scaled beat gesture frequency as a factor of scaled MIT scores (mean, below and above the mean) for the gesture-spontaneous and the gesture-encouraged condition of Group 1 and the gesture-encouraged condition of Group 2. A:** within-subject effects; **B:** between-subject effects.

who had MIT scores above the mean, the more beat gestures they produced, the better they did on the RAT, $\beta = 0.60$, $SE = 0.19$, $p < .001$. In GE2 (Group 2's gesture-encouraged condition), the only significant effect was the one for people with MIT scores below the mean–in their case, the more beat gestures they produced the worse they did on the RAT, $\beta = -0.48$, $SE = 0.15$, $p < .001$, (see Fig 2B).

*Representational gestures*: In the within-effects model including representational gestures, only the main effect of representational gesture frequency was significant (see Table 5), such that the more representational gestures participants produced, the better they performed on the RAT, irrespective of their MI skills or condition (see Fig 3A). In the between-effects model, there was a significant three-way interaction between representational gestures, group, and

**Table 5. Model summary for Hypothesis 2—Representational gestures.**

| | RAT | | | |
| --- | --- | --- | --- | --- |
| | within-subjects (Group 1) | | between-subjects (Group 1 & Group 2) | |
| | Coefficient | SE | Coefficient | SE |
| FIXED EFFECTS | | | | |
| Intercept | 1.50 | .93 | 1.23* | .53 |
| Representational Frequency | 2.55* | 1.13 | .78*** | .12 |
| Condition | .28 | 1.48 | | |
| Group | | | -.37 | .40 |
| MIT | -.11 | .77 | .50* | .21 |
| Representational*Condition | .63 | 2.22 | | |
| Representational*Group | | | -.19 | .24 |
| Representational*MIT | -1.58 | 1.11 | .41** | .13 |
| Condition*MIT | -2.32 | 1.58 | | |
| Group*MIT | | | .32 | .40 |
| Representational*Condition*MIT | -1.59 | 2.20 | | |
| Representational*Group*MIT | | | -.72** | .26 |
| | Variance | SD | Variance | SD |
| RANDOM EFFECTS | | | | |
| INTERCEPTS | | | | |
| Subject | 6.64 | 2.58 | 2.71 | 1.65 |
| Item | 5.77 | 2.40 | 4.79 | 2.19 |
| SLOPES | | | | |
| Condition|Subject | 11.68 | 3.42 | | |

*Note*: MIT–Mental Imagery Test.

Significance codes =

***$p < .001$

**$p < .01$

*$p < .05$, †$p < .1$.

MIT (see Table 3). Simple slope analyses revealed that representational gesture frequency positively and significantly predicted RAT scores in both groups, except for people with MI below the mean in Group 2 (see Fig 3B). Representational gesture frequency was a negative predictor of RAT scores for people with low MI in Group 2, however, this relationship was not significant, $\beta$ = -0.10, $SE$ = 0.17, $p$ = .57.

Hypothesis 3: People with more advanced mental imagery skills use their gestures more frequently in the RAT.

This hypothesis was tested only for the gesture-spontaneous condition of Group 1 (GS1) as we wanted to explore this relationship in a natural gesture condition. First, we fit a linear mixed-effects model where the overall gesture frequency was the outcome variable, MIT was a fixed effect, and subject and item were random intercepts. As MIT was not a significant predictor of overall gesture frequency (see Table 6), we did not test further gesture type models.

## vRAT (visual) within-group (GS1 and GE1) and between-group (GE1 and GE2) analyses

We ran the same models but this time with the vRAT scores instead of the RAT scores to test the same hypotheses for visual convergent thinking, i.e., vRAT performance.

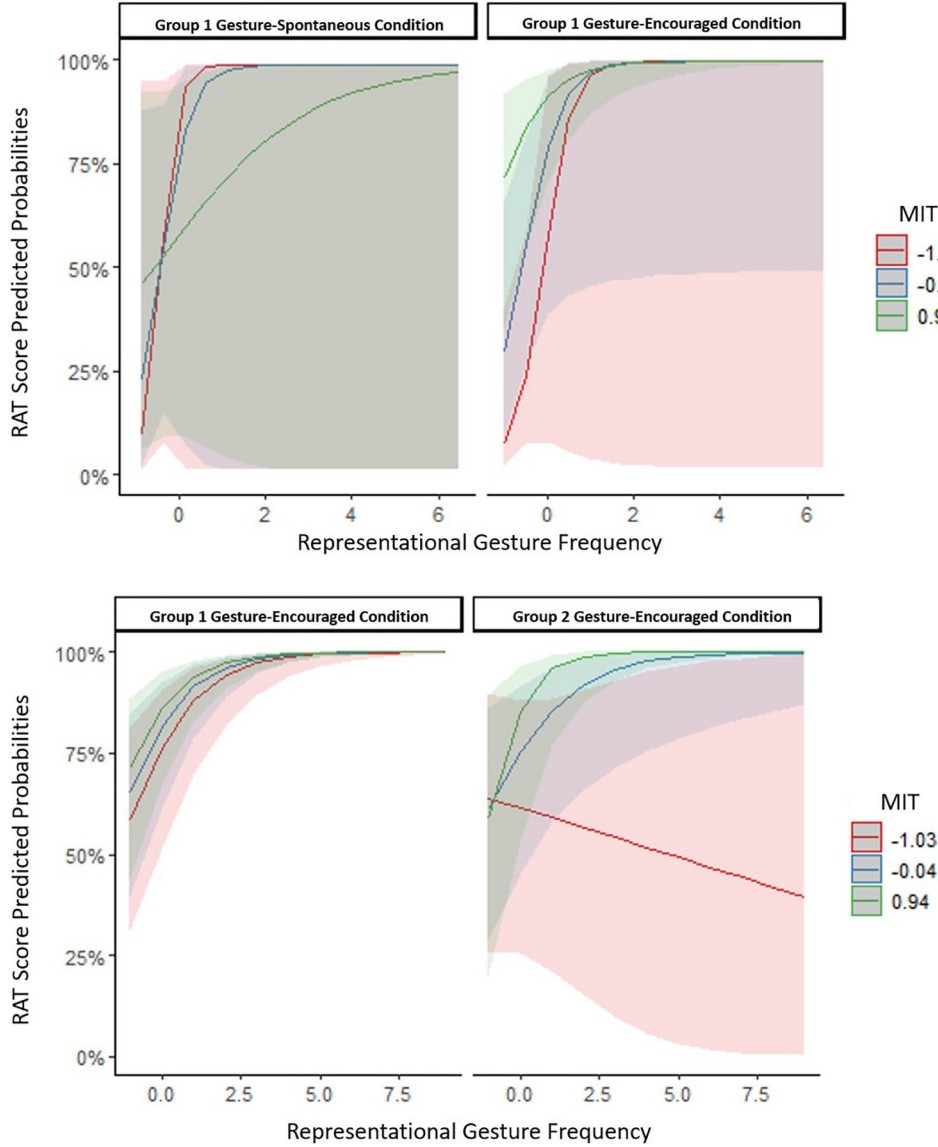

**Fig 3. The relationship between predicted probabilities of RAT scores and scaled representational gesture frequency as a factor of scaled MIT scores (mean, below and above the mean) for the gesture-spontaneous and the gesture-encouraged condition of Group 1 and the gesture-encouraged condition of Group 2. A:** within-subject effects; **B:** between-subject effects.

Hypothesis 1: People who naturally gesture more should do better on vRAT because of high mental imagery.

There were no significant main effects of total gesture frequency, or MIT scores, and the interaction between gestures and MIT was also not significant (see Table 2).

Hypothesis 2: Encouraging gestures improves (or hampers) vRAT performance and if it improves performance, this is specifically true for representational gestures.

Neither the within-subject nor the between-subject effects model was significant (see Table 2). Hence, gestures were not related to vRAT, or visual convergent thinking performance. Therefore, we did not further test any models for the specific gesture types.

Hypothesis 3: People with more advanced mental imagery skills use their gestures more frequently during the vRAT.

**Table 6. Summary of models for Hypothesis 3.**

| | RAT Overall Gesture Frequency | | vRAT Overall Gesture Frequency | |
|---|---|---|---|---|
| | Coefficient | SE | Coefficient | SE |
| FIXED EFFECTS | | | | |
| Intercept | -.30*** | .04 | -.24** | .07 |
| MIT | .03 | .04 | .04 | .07 |
| | Variance | SD | Variance | SD |
| RANDOM EFFECTS | | | | |
| INTERCEPTS | | | | |
| Subject | .05 | .23 | .16 | .40 |
| Item | .01 | .09 | .00 | .00 |

*Note*: MIT–Mental Imagery Test.

Significance codes =

***$p < .001$

**$p < .01$

*$p < .05$, †$p < .1$.

As with the RAT, this hypothesis was tested only for the gesture-spontaneous condition of Group 1 (GS1) so as to explore this relationship in a natural gesture condition. Since in the model with overall gesture frequency, MIT was not a significant predictor of gesture frequency (see Table 6), we did not further test the separate gesture types.

## Imagery styles

To test our hypothesis that RAT would be predicted by verbal and vRAT by object imagery styles we built different models for within- and between-effects.

**Verbal cognitive style and RAT.** The within-effects model for RAT had a significant three-way interaction between the verbal domain of OSIVQ, overall gesture frequency, and condition (see Table 7). In the GS condition, participants who identified as high verbalisers (a self-report score for verbal cognitive style above the mean), the more they gestured, the better they did on the RAT, $\beta = 4.83$, $SE = 2.18$, $p = .03$. Interestingly, in the GE condition of the same group (Group 1), it was the low (a self-report score for verbal cognitive style below the mean) and the average (a mean self-report score for verbal cognitive style) verbalisers who benefitted from gesturing, i.e., the more they gestured, the better they did on the RAT (for low verbalisers - $\beta = 3.01$, $SE = .63$, $p < .001$; for average verbalisers - $\beta = 1.45$, $SE = .39$, $p < .001$).

In the between-effects model, there was a significant interaction between verbal style and gesture frequency (see Table 7). Simple slope analysis showed that for people with mean, $\beta = .28$, $SE = .08$, $p < .001$, and above-the-mean verbal abilities, $\beta = .55$, $SE = .13$, $p < .001$, overall gesture frequency was a positive predictor of RAT scores.

**Spatial imagery and RAT.** The models for the spatial domain of OSIVQ were identical to the verbal domain ones and we only replaced the fixed effect of the verbal with the spatial scores. In the within-effects model (Group 1), there was a significant three-way interaction between spatial imagery, gesture frequency, and condition (see Table 8). Only when encouraged to gesture, participants with mean, $\beta = 1.00$, $SE = .34$, $p < .001$, and above-the-mean spatial imagery abilities, $\beta = 1.74$, $SE = .57$, $p < .001$, benefitted from gestures, i.e., there was a positive association between their total gesture frequency and RAT performance. In the between-effects model (Group 1 GE and Group 2 GE), there was a significant two-way

**Table 7. Model summary for verbal cognitive style.**

|  | RAT | | | | vRAT | | | |
|--|------|--|--|--|------|--|--|--|
|  | within-subjects (Group 1) | | between-subjects (Group1&Group2) | | within-subjects (Group 1) | | between-subjects (Group1&Group2) | |
|  | Coefficient | SE | Coefficient | SE | Coefficient | SE | Coefficient | SE |
| FIXED EFFECTS |  |  |  |  |  |  |  |  |
| Intercept | .63 | .62 | 1.25* | .54 | .84* | .37 | .74* | .38 |
| Gesture Frequency | 1.50*** | .45 | .27*** | .08 | .06 | .13 | .18 | .11 |
| Condition | -1.16 | .90 |  |  | -.09 | .20 |  |  |
| Group |  |  | -.25 | .39 |  |  | -.34 | .24 |
| Verbal | .20 | .53 | -.24 | .20 | .11 | .12 | -.04 | .13 |
| Gesture*Condition | .03 | .90 |  |  | .04 | .26 |  |  |
| Gesture*Group |  |  | .10 | .16 |  |  | .31 | .22 |
| Gesture*Verbal | .86 | .77 | .27*** | .08 | .01 | .09 | -.04 | .10 |
| Condition*Verbal | -.76 | 1.29 |  |  | .16 | .19 |  |  |
| Group*Verbal |  |  | -.31 | .39 |  |  | -.10 | .25 |
| Gesture*Condition*Verbal | 4.82** | 1.57 |  |  | .10 | .18 |  |  |
| Gesture*Group*Verbal |  |  | .26† | .15 |  |  | -.06 | .21 |
|  | Variance | SD | Variance | SD | Variance | SD | Variance | SD |
| RANDOM EFFECTS |  |  |  |  |  |  |  |  |
| INTERCEPTS |  |  |  |  |  |  |  |  |
| Subject | 10.36 | 3.22 | 2.61 | 1.62 | .33 | .58 | .49 | .70 |
| Item | 4.90 | 2.21 | 5.04 | 2.25 | 2.41 | 1.55 | 2.55 | 1.60 |
| SLOPES |  |  |  |  |  |  |  |  |
| Condition|Subject | 15.59 | 2.21 |  |  | .01 | .10 |  |  |

*Note*: Significance codes =

***$p < .001$

**$p < .01$

*$p < .05$

†$p < .1$.

interaction between spatial imagery and gesture frequency, (see Table 8). When encouraged to gesture, irrespective of which group they belonged to, people with below-the-mean spatial imagery abilities showed a negative association between gesture frequency and RAT scores, $\beta = -.41$, $SE = .15$, $p = .01$, while people with above-the-mean spatial imagery abilities exhibited a positive association between gesture frequency and RAT scores, $\beta = .50$, $SE = .11$, $p < .001$.

**Object imagery and RAT.** In the models built with the object imagery domain of OSIVQ, only gesture frequency was a positive predictor of RAT scores for the between-group effects (see Table 9). Irrespective of the group, or their object imagery abilities, the more participants gestured the better they did on the verbal convergent thinking task. In sum, object imagery was not associated with RAT performance.

**OSIVQ and vRAT.** We ran the same models for the vRAT as we did for the RAT. There were no significant main effects or interactions (see Tables 7–9).

To sum up, our hypothesis that RAT was associated with verbal cognitive style was confirmed, however, the hypothesis that vRAT was associated with object imagery was not.

**OSIVQ and MIT.** We ran correlational analyses (for Group 1 and Group 2 together) to test our hypothesis that MIT scores will be positively correlated with self-reported object and spatial imagery. There were no significant correlations between any of the OSIVQ domains

**Table 8. Model summary for spatial imagery.**

| | RAT | | | | vRAT | | | |
|---|---|---|---|---|---|---|---|---|
| | within-subjects (Group 1) | | between-subjects (Group1&Group2) | | within-subjects (Group 1) | | between-subjects (Group1&Group2) | |
| | Coefficient | SE | Coefficient | SE | Coefficient | SE | Coefficient | SE |
| FIXED EFFECTS | | | | | | | | |
| Intercept | .66 | .60 | 1.13* | .54 | .83* | .37 | .74* | .38 |
| Gesture Frequency | .85* | .38 | .03 | .08 | .02 | .11 | .19 | .11 |
| Condition | -.56 | .81 | | | -.09 | .19 | | |
| Group | | | -.57 | .40 | | | -.34† | .24 |
| Spatial | -.15 | .34 | .28 | .20 | .11 | .13 | .07 | .12 |
| Gesture*Condition | .29 | .76 | | | -.08 | .20 | | |
| Gesture*Group | | | -.33* | .17 | | | .35 | .23 |
| Gesture*Spatial | .10 | .34 | .27*** | .08 | -.12 | .14 | -.11 | .10 |
| Condition*Spatial | -.52 | .82 | | | .10 | .20 | | |
| Group*Spatial | | | .23 | .40 | | | .04 | .25 |
| Gesture*Condition*Spatial | -1.29* | .67 | | | .13 | .27 | | |
| Gesture*Group*Spatial | | | .29 | .19 | | | .13 | .20 |
| | Variance | SD | Variance | SD | Variance | SD | Variance | SD |
| RANDOM EFFECTS | | | | | | | | |
| INTERCEPTS | | | | | | | | |
| Subject | 7.25 | 2.69 | 2.74 | 1.65 | .31 | .56 | .48 | .69 |
| Item | 4.69 | 2.17 | 5.05 | 2.25 | 2.40 | 1.55 | 2.56 | 1.60 |
| SLOPES | | | | | | | | |
| Condition|Subject | 12.06 | 3.47 | | | .01 | .07 | | |

*Note*: Significance codes =

***$p < .001$

**$p < .01$

*$p < .05$

†$p < .1$.

and MIT (spatial imagery: $r$ (77) = .19, $p$ = .09; object imagery: $r$ (77) = -.09, $p$ = .41; verbal cognitive style: $r$ (77) = -.14, $p$ = .22).

## Discussion

The current study investigated the relationship between gesture production (spontaneous and encouraged) and creativity, focusing on verbal and visual convergent thinking. We also examined the individual differences in mental imagery skills and self-reported object, spatial, and verbal imagery skills in this relationship. There were three key findings. First, both spontaneous and encouraged gestures and mental imagery skills were associated with verbal but not visual convergent thinking performance. Second, encouraging gesture use did not in itself improve or hamper verbal convergent thinking, but the interaction between gesturing and one's imagery skills played a role on that relationship. Third, self-perceived verbal and spatial but not object imagery skills were associated with the effects of gestures on verbal convergent thinking.

### Gestures' role in verbal and visual convergent thinking with mental imagery

When people with average and above-average mental imagery skills gestured more during the verbal convergent thinking task (RAT), both spontaneously and when encouraged, they

**Table 9. Model summary for object imagery.**

| | RAT | | | | vRAT | | | |
|---|---|---|---|---|---|---|---|---|
| | within-subjects (Group 1) | | between-subjects (Group1&Group2) | | within-subjects (Group 1) | | between-subjects (Group1&Group2) | |
| | Coefficient | SE | Coefficient | SE | Coefficient | SE | Coefficient | SE |
| FIXED EFFECTS | | | | | | | | |
| Intercept | .46 | .61 | 1.20* | .54 | .82* | .37 | .74* | .38 |
| Gesture Frequency | .61 | .39 | .19* | .08 | .03 | .10 | .19† | .11 |
| Condition | -.89 | .87 | | | -.10 | .19 | | |
| Group | | | -.35 | .40 | | | -.34 | .24 |
| Object | -.22 | .36 | -.01 | .20 | .11 | .13 | .15 | .12 |
| Gesture*Condition | -.77 | .79 | | | -.07 | .19 | | |
| Gesture*Group | | | -.08 | .15 | | | .35 | .22 |
| Gesture*Object | .72† | .41 | -.03 | .08 | -.00 | .14 | .03 | .12 |
| Condition*Object | 1.13 | .87 | | | .04 | .21 | | |
| Group*Object | | | -.08 | .39 | | | -.13 | .24 |
| Gesture*Condition*Object | .40 | .81 | | | -.24 | .28 | | |
| Gesture*Group*Object | | | -.02 | .15 | | | -.18 | .24 |
| | Variance | SD | Variance | SD | Variance | SD | Variance | SD |
| RANDOM EFFECTS | | | | | | | | |
| INTERCEPTS | | | | | | | | |
| Subject | 7.40 | 2.72 | 2.71 | 1.65 | .29 | .54 | .45 | .67 |
| Item | 4.94 | 2.22 | 5.02 | 2.24 | 2.41 | 1.55 | 2.52 | 1.59 |
| SLOPES | | | | | | | | |
| Condition\|Subject | 14.56 | 3.82 | | | .01 | .08 | | |

*Note*: Significance codes =

***$p < .001$

**$p < .01$

*$p < .05$

†$p < .1$.

performed better than people who used fewer gestures in the same task. On the other hand, when people with lower mental imagery skills gestured, both spontaneously and when encouraged, the more they gestured the worse they did on the verbal convergent thinking task. As mental imagery skills were not a significant predictor of gesture frequency, it means that people with high imagery skills do not necessarily gesture more when solving verbal convergent thinking problems, but unlike people with low imagery, gesturing helps them successfully solve those problems. This finding highlights the importance of imagery abilities for benefitting from gestures for verbal convergent thinking.

We then tested the hypothesis that representational gestures might have been more beneficial compared to beat gestures, as representational gestures are directly related to the content of one's mental images [11–13]. Representational gestures were indeed a positive predictor of verbal convergent thinking for participants in both conditions (spontaneous and encouraged) and groups, except for those with low mental imagery skills in Group 2 (the group that attended a gesture-encouraged condition only). As participants in this group had not had previous experience with the task, having low mental imagery while trying to gesture when solving novel verbal convergent thinking problems might have led to increased cognitive load and

hence, decreased task performance. These findings complement previous research showing that representational gestures benefit spatial problem solving, both for adults and children [33,81]. However, when participants have low spatial working memory, which corresponds to having low mental imagery capacity in our study, they do not benefit from gesturing [82].

Regarding beat gestures, we found that for people with low and average mental imagery skills, producing beat gestures in their encouraged gesture condition hampered convergent thinking. Even though we did not test working memory capacity, previous research shows that working memory and mental imagery skills are positively correlated [60], thus, it is possible that for people with low mental imagery skills being encouraged to gesture might have increased their working memory load and hence, decreased RAT performance. This assumption is supported by previous evidence that gesturing during an analogical verbal reasoning task can be detrimental to memory recall due to increased cognitive load [73]. On the other hand, for people with high mental imagery skills, when they have been previously exposed to similar RAT trials (in the gesture-spontaneous condition), the more beat gestures they produced when their gestures were encouraged, the better they performed on the task. Hence, high mental imagery skills might modulate the positive relationship between beat gestures and convergent thinking when people have also had some previous experience with the task. This pattern of results is consistent with the findings that beat gestures can benefit word retrieval and memory recall for adults (e.g., [42,43]). One interpretation could be that performing beat gestures is less cognitively taxing than performing representational gestures, where one might need to consciously adjust the shape of their hands according to the content of speech when encouraged to gesture. As beat gestures do not bear any semantic information, they could be easier to perform. The finding that beat gestures are helpful only when participants have had some previous experience with the task is supported by previous research by Cho and So [83]. In that study, gesturing during a mental abacus task was only helpful for participants who had a certain amount of experience with the task but not for those who were just getting familiar with the task. Interestingly, gesturing did not benefit their mental abacus performance if they had had a lot of experience with the task. Even though the RAT is not a mental manipulation task, a venue for further investigation would be to also test a group that has had more experience with the task, for example, attending additional 10 trials, and test if beat gestures would still be beneficial for their performance.

We did not find any effects of gestures or mental imagery in the visual convergent thinking task (vRAT). The reason why gestures were associated with verbal, but not visual convergent thinking may be explained by the idea that gestures are used to activate and maintain visuospatial imagery [10–13]. In the visual task, participants had the objects displayed on the screen in front of them (i.e., the images of the objects were accessible), so they did not have to be retrieved or activated in participants' minds. However, in the verbal task, participants were presented with the words. Gestures, in addition to good imagery skills, might have helped them activate the images corresponding to the words, which in turn enhanced their task performance.

In sum, spontaneous and encouraged overall gesture use helps those with average and high mental imagery skills as they perform better on a verbal convergent thinking task. Yet, it hampers the performance of those with low mental imagery skills. More interestingly, we see a "dissociation" between beat and representational gestures. While beat gestures hamper verbal convergent thinking in people with lower mental imagery capacity and help those with high imagery when they have had experience with the task, representational gestures improve verbal convergent thinking for everyone apart from people with low mental imagery who have not had experience with the task.

## Imagery styles, gestures, and convergent thinking

We also extended our findings on the effects of mental imagery, tested by a standardised test (MIT), on gestures and convergent thinking, by adding an additional self-report measure of imagery styles. This allowed us to consider the multi-faceted nature of imagery as previous research provides substantial evidence for a distinction between *visual-object* and *visual-spatial* imagery and *verbal-analytical* thinking [64–66]. Even though we hypothesised that object imagery should be related to visual convergent thinking, we did not find any effects of imagery styles for visual convergent thinking. As mentioned above, solving the visual RAT did not require any imagery skills as the objects of the stimuli were presented on a screen in front of the participants. However, future studies can test this hypothesis by showing the objects for a limited amount of time, then removing them from sight and allowing participants to think about the answer while retrieving the objects from their short-term memory. Then, object imagery might play a role in visual convergent thinking. Furthermore, object imagery has been linked to artistic creativity [66], which has been found to be related more to divergent rather than convergent thinking [84]. As convergent thinking is conventionally measured by the verbal RAT, further research should investigate if there is any relation between visual convergent thinking, e.g., the visual RAT, and artistic creativity.

We found an interaction between the level of verbal skills and the amount at which gestures contribute to verbal convergent problem-solving. In general, for people with better verbal skills, gesturing was associated with better RAT performance, however, when familiar with the task, even those with lower verbal skills benefitted from gesturing. Interestingly, there was also an effect of self-reported spatial imagery and gestures on verbal convergent thinking and this effect was analogous to the results we obtained for gestures and the standardised mental imagery test. In particular, for people with high spatial imagery skills, encouraging their gestures improves their verbal convergent thinking performance, however, for people with low spatial imagery, encouraging gesture use has a negative effect on verbal convergent thinking. Even though the positive correlation between the spatial domain of imagery and MIT was not statistically significant, it was the only positive one among the three domains of the OSIVQ that was also approaching significance ($p = .09$), which together with the analogous findings mentioned above, points to the inference that MIT is a measure of the spatial domain of visual imagery. Last, object imagery was not related to verbal convergent thinking. A future study should address the interaction between gesture types and perceived imagery styles and their effects on convergent thinking.

Our findings regarding perceived imagery styles are in line with previous literature by adding further evidence for the nuances expected in the intricate relationship between gestures and thinking. For example, people may benefit more from observing gestures if they have high spatial abilities but benefit more from speech if they have high verbal abilities (for a review, see [71]). In our study, we showed that for gesture production, even people with lower verbal abilities can benefit from gesturing for verbal problem-solving, however, high spatial imagery abilities are necessary so that gestures can be efficient in verbal problem-solving. Previous research shows that young adults may use *gesture-as-a-compensation-tool* for lower verbal and spatial skills [71]. The reason why in our study gestures could not compensate for low spatial resources could be due to the nature of our task–it required verbal rather than spatial knowledge. However, high spatial skills still contributed to task performance, which might have to do with the mechanisms of gesture production. We execute gestures in the immediate space around us and being aware of how to use that space successfully might help us produce the right gestures to benefit verbal problem-solving. It would be interesting to test if gestures can compensate for low spatial imagery in a non-verbal spatial creativity task that is related to

mentally manipulating objects in imagery to construct a new original object, e.g., the Creative Synthesis Task [85].

Last, our findings are both similar and contrasting to previous research showing that verbal creativity is mostly domain-specific but also sensitive to processes in the visual domain, and that visual creativity is only related to visual abilities [86]. Our results are contradictory to this finding as we did not find any association between visual creativity and object or spatial imagery. However, it could be due to the nature of our task, which did not require creating a novel object, figure, or drawing. Future research can examine the relationship among gestures, imagery, and figural creativity. Our results supported the findings of Palmiero et al. [86] that verbal creativity is related to both verbal and visual abilities as we found a moderating effect of verbal and spatial skills on the effect of gestures on verbal problem-solving. Interestingly, object imagery did not play a role in that relationship, which needs to be further investigated using different creativity tasks.

Although the present study presents supporting evidence for the role of mental imagery in the gesture-creativity relationship, we do not know whether participants used their imagery while solving the convergent thinking problems. Future research could control for this by using a binocular rivalry task, which interferes with visual imagery if the latter is being used while solving the main task. Research shows that individuals with poor visual imagery are not affected by the background luminance of binocular rivalry and perform above chance on visual working memory, using different strategies, e.g., verbal ones [87].

If, as the present study suggests, the degree to which people benefit from gestures for convergent thinking depends on their mental imagery skills and styles, then there is also a need for research that explores the interaction between gestures and imagery for another key component of creative thought–*divergent thinking*, or the ability to generate many different and novel ideas for a problem. Even though a meta-analysis of the imagery-creativity relationship reported that imagery accounts for a very limited amount of the variance in both verbal and figural divergent thinking [88], the role of gestures, which have the potential to reinforce imagery, is understudied and much work remains to be done before a full understanding of the gesture-imagery-creativity interplay is established.

Last, one of the main challenges both in creativity assessment and gesture coding can be achieving high interrater reliability, mainly due to subjectivity in human ratings. Although convergent thinking measurement is immune to human bias because correct responses are usually predetermined, classical divergent thinking measures, such as the AUT, are prone to rater subjectivity. These limitations are currently being addressed by automation of creativity assessment with computational methods that use natural language processing [89,90]. Moreover, newly emerging technological improvements, such as machine-learning algorithms also offer a solution to the challenges in gesture coding (e.g., recently proposed state-of-the-art gesture recognition methods [91–94]). Even though these studies used large datasets to train their models, there is still a big diversity in how people use their gestures, which makes it difficult to train machines to detect and classify gestures. Alternatively, human raters benefit from other verbal and nonverbal cues, such as speech, prosody, and facial expressions, to identify and classify hand gestures. Even if training machines for gesture recognition could be an arduous task, it is a promising path for valuable interdisciplinary work between the fields of psycholinguistics and computer science.

## Conclusion

The present study examined the role of hand gestures in convergent thinking, providing further evidence for the growing body of literature not only on embodied creativity but also on

the individual cognitive differences in gesture production. We showed that gestures can be a good candidate for enhancing verbal convergent thinking when one has good mental and spatial imagery skills, but gestures can also compensate for low verbal skills. Moreover, the types of gestures executed would matter. While representational gestures help almost everybody regardless of their imagery skills, beat gestures benefit only those with high imagery skills and a certain amount of experience with the task at hand. Therefore, the practical implications of the current findings highlight the importance of considering *when* and *who* should be encouraged to use gestures when solving creative problems. To have a complete picture of the gesture-creativity interplay, future research should address the role of gestures in figural creativity and divergent thinking.

## Supporting information

**S1 File. Supporting information.** A file including a detailed description and visualisation of gesture types and frequencies across groups and conditions and the linear mixed-effects modelling results for the individual representational gesture types, i.e., iconic, metaphoric and pointing/deictic gestures, and palm-revealing gestures.
(DOCX)

## Acknowledgments

The authors sincerely thank Melek Öyküm Yalçın, Dila Gürer, İrem Türkmen, Sarp Özdemir, Helin Erden, and Ayşe Başak Ersoy for assistance with data collection, gesture coding, and data entry; Demet Özer, Dilay Karadöller, Emir Akbuğa, and Burcu Arslan for their help in data analyses and gesture coding, the members at the Language & Cognition Lab for their valuable feedback; and all the participants for their time and effort.

## Author Contributions

**Conceptualization:** Gyulten Hyusein, Tilbe Göksun.

**Data curation:** Gyulten Hyusein.

**Formal analysis:** Gyulten Hyusein.

**Funding acquisition:** Tilbe Göksun.

**Investigation:** Gyulten Hyusein.

**Methodology:** Gyulten Hyusein, Tilbe Göksun.

**Project administration:** Gyulten Hyusein.

**Resources:** Tilbe Göksun.

**Software:** Gyulten Hyusein.

**Supervision:** Tilbe Göksun.

**Validation:** Tilbe Göksun.

**Visualization:** Gyulten Hyusein.

**Writing – original draft:** Gyulten Hyusein.

**Writing – review & editing:** Gyulten Hyusein, Tilbe Göksun.

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
