## [Decision Letter · Decision Letter 0]

28 Oct 2022

PONE-D-22-19099The creative interplay between hand gestures, convergent thinking, and mental imageryPLOS ONE

Dear Dr. Hyusein,

Thank you for submitting your manuscript to PLOS ONE. After careful consideration, we feel that it has merit but does not fully meet PLOS ONE’s publication criteria as it currently stands. Therefore, we invite you to submit a revised version of the manuscript that addresses the points raised during the review process.

After a careful evaluation of two reviewers and in a full respect for any minor concerns that might have emerged during the revision,particularly regarding the comments of reviewer 1 ,please take into consideration these two minor revisions :1. Make a more comprehensive line with  shortening sentences and/or consider revising it (last paragraph, page 7,line 195);2. Please clarify the discrepancy between the sample written in the abstract section and those who are revealed in the results section (line 415);there is an unclear procedure between the 78 and/or 80 tested  participants. Please give an explanation if there were any changes  during the collection of the final valid number of participants.3. Please also consider to give a better resolution of the figures and identify the vertical axis and plotted scores in order to make them clearer for the public reader.Please note that I have also acted as a reviewer for your work under "Reviewer 2" comments. Both reviewers have found your work really interesting and recommended a positive evaluation based on the overall structure and scientific operationalization on the whole work,but,you are pleased to consider and revise the above concerns in order to have a ready-to-publish article at the most convenient time.You are pleased to submit your revised manuscript by 8th November 2022. If you will need more time than this to complete your revisions, please reply to this message or contact the journal office at plosone@plos.org. Please include the following items when submitting your revised manuscript:A rebuttal letter that responds to each point raised by the academic editor and reviewer(s). You should upload this letter as a separate file labeled 'Response to Reviewers'.A marked-up copy of your manuscript that highlights changes made to the original version. You should upload this as a separate file labeled 'Revised Manuscript with Track Changes'.An unmarked version of your revised paper without tracked changes. You should upload this as a separate file labeled 'Manuscript'.We look forward to receiving your revised manuscript.

Kind regards,

Dr. Silva Ibrahimi, PhD

Academic Editor

PLOS ONE

Journal Requirements:

Additional Editor Comments (if provided):

Reviewers' comments:

Reviewer's Responses to Questions

**Comments to the Author**

1. Is the manuscript technically sound, and do the data support the conclusions?

Reviewer #1: Yes

Reviewer #2: Yes

2. Has the statistical analysis been performed appropriately and rigorously? 

Reviewer #1: Yes

Reviewer #2: Yes

3. Have the authors made all data underlying the findings in their manuscript fully available?

Reviewer #1: Yes

Reviewer #2: Yes

4. Is the manuscript presented in an intelligible fashion and written in standard English?

Reviewer #1: Yes

Reviewer #2: Yes

5. Review Comments to the Author

Reviewer #1: The target paper presents and discusses the results of one study aiming to clarify the relationship between hand gestures, mental imagery, and convergent thinking. The study is timely and of relevance for, even thought several papers disclose a relation between hand gesturing and divergent thinking, no study to date extended these findings to convergent thinking.

Overall, the manuscript is well written and clear. The state of the art is particularly clear and thorough, but also straightforward in clarifying the hypothesis and its rational. The design of the study is clever (particularly on the manipulation of the variables both within and between subjects), the analyses appropriate and the results clear.

For all the above, I endorse this manuscript to be published. I have but a few minor points and suggestions.

1. The first sentence in the last paragraph on page 7 (staring on line 195) is a bit awkward to read and tries to squeeze information which would be easier to follow if presented in two or three separate sentences.

2. First paragraph of the results section (line 415). This information would most naturally fit in the participants section. Also, it goes against what is stated in that section, which reads "Therefore, the data of 80 participants were included in the final analysis". Likewise, shouldn't the abstract read that 78, instead of 80, participants were tested?

3. The figures seemed to me to be of low quality. Also, it would be preferable if the vertical axis specified, in each image, what score is being plotted (as they are, all images just read "score" and the reader has to refer to the caption or the legend). Finally, the labels of the plots (A and B) should be presented in the top left corner of each plot, instead of in the bottom left corner.

Reviewer #2: This is a very interesting research! Although the efforts to study the dynamics of convergent thinking are known in literature,the present study gives a fresh and comprehensive value in determining associations and potential linkage between gesticulation,mental representation and mental imagery. Further research areas are also properly addressed as the present work is a country located-focused.

6. PLOS authors have the option to publish the peer review history of their article (what does this mean?). If published, this will include your full peer review and any attached files.

Reviewer #1: **Yes: **Nuno De Sá Teixeira

Reviewer #2: No

---

## [Author Response · Author response to Decision Letter 0]

9 Nov 2022

Response to Reviewers

Reviewer 1: The target paper presents and discusses the results of one study aiming to clarify the relationship between hand gestures, mental imagery, and convergent thinking. The study is timely and of relevance for, even though several papers disclose a relation between hand gesturing and divergent thinking, no study to date extended these findings to convergent thinking.

Overall, the manuscript is well written and clear. The state of the art is particularly clear and thorough, but also straightforward in clarifying the hypothesis and its rational. The design of the study is clever (particularly on the manipulation of the variables both within and between subjects), the analyses appropriate and the results clear.

For all the above, I endorse this manuscript to be published. I have but a few minor points and suggestions.

Response: We thank the reviewer for their positive feedback and helpful comments on the paper.

Reviewer 1: The first sentence in the last paragraph on page 7 (staring on line 195) is a bit awkward to read and tries to squeeze information which would be easier to follow if presented in two or three separate sentences.

Response: We have revised the sentence by splitting it into two separate sentences and paraphrasing it as follows: “There have not been any studies investigating gestures’ effects on convergent thinking. However, research has shown that fluid arm movements, squeezing a ball, or enacting metaphors with hands could improve convergent thinking [1, 2, 46, 47].”

Reviewer 1: First paragraph of the results section (line 415). This information would most naturally fit in the participants section. Also, it goes against what is stated in that section, which reads "Therefore, the data of 80 participants were included in the final analysis". Likewise, shouldn't the abstract read that 78, instead of 80, participants were tested?

Response: We have moved the first paragraph of the Results section to the Participants subsection of the Method section (p.12, line 321) explaining the reason for the discrepancy in the two sample sizes. We have also included the descriptive statistics (mean age, standard deviation, and the number of females) of the RAT sample. The newly added paragraph is as follows: 

“For the RAT analyses, two participants were later excluded because one of them was not shown the RAT triads on the screen, but the experimenter only read them out loud; and the second one was excluded because they were an outlier due to previous research familiarity with the task. Thus, our final sample consisted of 78 participants (Mage = 21.3, SD = 2.47; 50 females) for the RAT analyses and 80 participants (Mage = 21.3, SD = 2.46; 52 females) for the vRAT analyses.”We have also removed the number of the sample size from the abstract, so it is not confusing to the reader: 

“We tested young adults on verbal and visual convergent thinking, controlling for their mental imagery skills.”

Reviewer 1: The figures seemed to me to be of low quality. Also, it would be preferable if the vertical axis specified, in each image, what score is being plotted (as they are, all images just read "score" and the reader has to refer to the caption or the legend). Finally, the labels of the plots (A and B) should be presented in the top left corner of each plot, instead of in the bottom left corner.

Response: We have replaced the initial figures with higher-quality ones. We have changed the names of the vertical axes from “Score” to “RAT Score Predicted Probabilities.” Labels (A and B) have been moved to the upper left corners. We have replaced the grey background and grid lines with a white background to make the slopes easier to read. We have changed the acronyms (GS1, GE1, and GE2) in the graphs with the full names of the groups and conditions since the space allows, and again, it makes the graphs easier to read and grasp. Finally, we removed the acronyms from the captions as they are no longer needed.

Reviewer 2: This is a very interesting research! Although the efforts to study the dynamics of convergent thinking are known in literature, the present study gives a fresh and comprehensive value in determining associations and potential linkage between gesticulation, mental representation and mental imagery. Further research areas are also properly addressed as the present work is a country located-focused.

Response: We thank the reviewer for their thoughtful and encouraging comments.

---

## [Decision Letter · Decision Letter 1]

7 Feb 2023

PONE-D-22-19099R1The creative interplay between hand gestures, convergent thinking, and mental imageryPLOS ONE

Dear 

Dear Dr. Hyusein,

Thank you for submitting your manuscript to PLOS ONE. After careful consideration, we feel that it has merit but does not fully meet PLOS ONE’s publication criteria as it currently stands. Therefore, we invite you to submit a revised version of the manuscript that addresses the points raised during the review process.

Your revised work has generally addressed all the previous suggestions and recommendations; it is well-structured and technically sound. The only concern to kindly be considered by you and in respecting our valuable reviewer's opinion regards the suggestions of Reviewer 3 to cite some more works within the micro-gesture recognition analysis. You can find the reviewer's suggestion below.

Please kindly consider addressing the suggestion  by Mar 24 2023 11:59PM

If you will need more time than this to complete your revisions, please reply to this message or contact the journal office at plosone@plos.org. Please include the following items when submitting your revised manuscript:A rebuttal letter that responds to each point raised by the academic editor and reviewer(s). You should upload this letter as a separate file labeled 'Response to Reviewers'.A marked-up copy of your manuscript that highlights changes made to the original version. You should upload this as a separate file labeled 'Revised Manuscript with Track Changes'.An unmarked version of your revised paper without tracked changes. You should upload this as a separate file labeled 'Manuscript'.We look forward to receiving your revised manuscript.

Kind regards,

Silva Ibrahimi, PhD

Academic Editor

PLOS ONE

Journal Requirements:

Reviewers' comments:

Reviewer's Responses to Questions

**Comments to the Author**

1. If the authors have adequately addressed your comments raised in a previous round of review and you feel that this manuscript is now acceptable for publication, you may indicate that here to bypass the “Comments to the Author” section, enter your conflict of interest statement in the “Confidential to Editor” section, and submit your "Accept" recommendation.

Reviewer #1: All comments have been addressed

Reviewer #3: All comments have been addressed

2. Is the manuscript technically sound, and do the data support the conclusions?

Reviewer #1: Yes

Reviewer #3: Yes

3. Has the statistical analysis been performed appropriately and rigorously? 

Reviewer #1: Yes

Reviewer #3: Yes

4. Have the authors made all data underlying the findings in their manuscript fully available?

Reviewer #1: Yes

Reviewer #3: Yes

5. Is the manuscript presented in an intelligible fashion and written in standard English?

Reviewer #1: Yes

Reviewer #3: Yes

6. Review Comments to the Author

Reviewer #1: The authors adequately addressed all my comments and the manuscript is considerably improved. In particular, the new figures are much easier to read and more clearly convey the found outcomes.

Reviewer #3: The authors kindly revised the papers according to the comments. Some gesture recognition related papers should be cited.

[1] X. Liu, H. Shi, H. Chen, Z. Yu, X. Li, and G. Zhao. “iMiGUE: An Identity-free Video Dataset for Micro-Gesture Understanding and Emotion Analysis,” IEEE/CVF conference on computer vision and pattern recognition, pp. 10631-10642, 2021.

[2] X. Liu, H. Shi, X. Hong, H. Chen, D. Tao, and G. Zhao “3D Skeletal Gesture Recognition via Hidden States Exploration,” IEEE Transactions on Image Processing, Vol. 29, pp. 4583–4597,2020.

[3] Z. Yu, B. Zhou, J. Wan, P. Wang, H. Chen, X. Liu, S. Li, and G. Zhao. “Searching Multi-Rate and Multi-Modal Temporal Enhanced Networks for Gesture Recognition,” IEEE Transactions on Image Processing, 2021.

[4] X. Liu and G. Zhao, “3D Skeletal Gesture Recognition using Sparse Coding of Time-Warping Invariant Riemannian Trajectories,” IEEE Transactions on Multimedia, Vol. 23, pp. 1841–1854, 2021.

7. PLOS authors have the option to publish the peer review history of their article (what does this mean?). If published, this will include your full peer review and any attached files.

Reviewer #1: **Yes: **Nuno Alexandre De Sá Teixeira

Reviewer #3: No

---

## [Author Response · Author response to Decision Letter 1]

24 Feb 2023

Response to Reviewers

Reviewer 1: The authors adequately addressed all my comments and the manuscript is considerably improved. In particular, the new figures are much easier to read and more clearly convey the found outcomes.

Response: We thank the reviewer for their original comment and also believe that their suggestion has improved the quality of the figures and the paper overall.

Reviewer 3: The authors kindly revised the papers according to the comments. Some gesture recognition related papers should be cited.

[1] X. Liu, H. Shi, H. Chen, Z. Yu, X. Li, and G. Zhao. “iMiGUE: An Identity-free Video Dataset for Micro-Gesture Understanding and Emotion Analysis,” IEEE/CVF conference on computer vision and pattern recognition, pp. 10631-10642, 2021.

[2] X. Liu, H. Shi, X. Hong, H. Chen, D. Tao, and G. Zhao “3D Skeletal Gesture Recognition via Hidden States Exploration,” IEEE Transactions on Image Processing, Vol. 29, pp. 4583–4597,2020.

[3] Z. Yu, B. Zhou, J. Wan, P. Wang, H. Chen, X. Liu, S. Li, and G. Zhao. “Searching Multi-Rate and Multi-Modal Temporal Enhanced Networks for Gesture Recognition,” IEEE Transactions on Image Processing, 2021.

[4] X. Liu and G. Zhao, “3D Skeletal Gesture Recognition using Sparse Coding of Time-Warping Invariant Riemannian Trajectories,” IEEE Transactions on Multimedia, Vol. 23, pp. 1841–1854, 2021.

Response: We thank the reviewer for their suggested readings. We addressed them in an additional paragraph that we included in the Discussion section (line 808-822) and updated the reference list accordingly (line 1076-1093). As the suggested papers were not very relevant to our claims, we did not go into much detail when discussing them. 

The newly added paragraph is as follows: “Last, one of the main challenges both in creativity assessment and gesture coding can be achieving high interrater reliability, mainly due to subjectivity in human ratings. Although convergent thinking measurement is immune to human bias because correct responses are usually predetermined, classical divergent thinking measures, such as the AUT, are prone to rater subjectivity. These limitations are currently being addressed by automation of creativity assessment with computational methods that use natural language processing [89, 90]. Moreover, newly emerging technological improvements, such as machine-learning algorithms also offer a solution to the challenges in gesture coding (e.g., recently proposed state-of-the-art gesture recognition methods [91-94]). Even though these studies used large datasets to train their models, there is still a big diversity in how people use their gestures, which makes it difficult to train machines to detect and classify gestures. Alternatively, human raters benefit from other verbal and nonverbal cues, such as speech, prosody, and facial expressions, to identify and classify hand gestures. Even if training machines for gesture recognition could be an arduous task, it is a promising path for valuable interdisciplinary work between the fields of psycholinguistics and computer sciences.”

---

## [Decision Letter · Decision Letter 2]

20 Mar 2023

The creative interplay between hand gestures, convergent thinking, and mental imagery

PONE-D-22-19099R2

Dear Author,

We’re pleased to inform you that your manuscript has been judged scientifically suitable for publication and will be formally accepted for publication once it meets all outstanding technical requirements.

Kind regards,

Silva Ibrahimi, PhD

Academic Editor

PLOS ONE

Additional Editor Comments (optional):

Reviewers' comments:

Reviewer's Responses to Questions

**Comments to the Author**

1. If the authors have adequately addressed your comments raised in a previous round of review and you feel that this manuscript is now acceptable for publication, you may indicate that here to bypass the “Comments to the Author” section, enter your conflict of interest statement in the “Confidential to Editor” section, and submit your "Accept" recommendation.

Reviewer #1: All comments have been addressed

Reviewer #3: All comments have been addressed

2. Is the manuscript technically sound, and do the data support the conclusions?

Reviewer #1: Yes

Reviewer #3: Yes

3. Has the statistical analysis been performed appropriately and rigorously? 

Reviewer #1: Yes

Reviewer #3: Yes

4. Have the authors made all data underlying the findings in their manuscript fully available?

Reviewer #1: Yes

Reviewer #3: Yes

5. Is the manuscript presented in an intelligible fashion and written in standard English?

Reviewer #1: Yes

Reviewer #3: Yes

6. Review Comments to the Author

Reviewer #1: The authors addressed all remaining issues and I reckon the manuscript is now ready to be accepted for publication

Reviewer #3: No further comments. The authors have effectively addressed all my concerns and I recommend accepting the manuscript.

7. PLOS authors have the option to publish the peer review history of their article (what does this mean?). If published, this will include your full peer review and any attached files.

Reviewer #1: **Yes: **Nuno Alexandre De Sá Teixeira

Reviewer #3: No

---

## [Editor Report · Acceptance letter]

29 Mar 2023

PONE-D-22-19099R2 

The creative interplay between hand gestures, convergent thinking, and mental imagery 

Dear Dr. Hyusein:

I'm pleased to inform you that your manuscript has been deemed suitable for publication in PLOS ONE. Congratulations! Your manuscript is now with our production department. 

Kind regards, 

on behalf of

Dr. Silva Ibrahimi 

Academic Editor

PLOS ONE